# Impact of Drinking Water Source and Sanitation Facility on Malnutrition Prevalence in Children under Three: A Gender-Disaggregated Analysis Using PDHS 2017–18

**DOI:** 10.3390/children9111674

**Published:** 2022-10-31

**Authors:** Rafit Saheed, Muhammad Shahid, Jun Wang, Madeeha Gohar Qureshi, Xiaoke Sun, Asma Bibi, Sidra Zia, Kun Tang

**Affiliations:** 1School of Public Policy, Pakistan Institute of Development Economics (PIDE), Islamabad 44000, Pakistan; 2World Health Organization, Sub-Office, Peshawar 25000, Pakistan; 3School of Insurance and Economics, University of International Business and Economics (UIBE), Beijing 100029, China; 4Vanke School of Public Health, Tsinghua University, Beijing 100084, China; 5Department of Economics, Pakistan Institute of Development Economics (PIDE), Islamabad 44000, Pakistan; 6Department of Risk Management and Insurance, School of Finance, Guangdong University of Foreign Studies, Guangzhou 510006, China; 7Independent researcher in Applied Psychology, Lahore 54000, Pakistan

**Keywords:** gender analysis, malnutrition, PDHS, sanitation facility, water source

## Abstract

Objectives: The proposed research studied the determinants of male and female child malnutrition in Pakistan. More specifically, it observed the role of the sanitation facility and drinking water source as important determinants of malnutrition in a gender analysis. Methods: Novel data of 1010 children under three years of age from PDHS 2017–18 were used. A CIAF (Cumulative Index for Anthropometric Failure) was established to assess malnourishment in the children. Discrete-choice logistic methodology was applied in this empirical research to study the likelihood of malnourishment in children. Results: The logistic regression results depicted that factors such as a child belonging to a deprived area, the status of home wealth, and the education of the mother were common determinants of malnutrition in children. Factors such as a child having diarrhea (OR = 1.55, CI = 0.96–2.50) and the drinking water source (OR = 0.62, CI = 0.37–1.03) were separate prominent predictors of malnutrition in male children whereas the sanitation facility was the main determinant of malnutrition in female children (OR = 0.64, CI = 0.43–0.95). Conclusion: This study concludes that important links exist between the drinking water source and male child malnutrition and between sanitation facilities and female child malnutrition.

## 1. Introduction

Worldwide, South Asia has a maximum prevalence of stunting at 26.9 million (38.9%) [1]. The Pakistan Demographic and Health Survey (PDHS) 2017–2018 reported that 38% of children were stunted, 23% were underweight, and 8% were wasted [2,3]. Studies have shown that male children are preferred by their parents and girls are more socio-economically deprived than boys. Gender discrimination at home makes girls more malnourished than male children [4,5,6]. However, a few researchers have argued that male children are at higher levels of malnourishment compared with their female counterparts due to a poor socio-economic status of the household [7,8,9]. Undernourished girls in their adult life produce malnourished babies and a vicious cycle of undernutrition and poverty continues [8].

This study focused on the gender-based determinants of child malnutrition, with a greater emphasis on water and sanitation as important factors of malnutrition in children under three in Pakistan. In developing countries, the most common phenomena are a sex bias or gender preferences, or discrimination in poor households [10]. Unfortunately, in many poor families, sons are preferred over daughters [10]. The main reason is that males are considered to be breadwinners whereas females are perceived to be a burden on a poor family [10]. This preference compels many poor parents to discriminate in terms of food, education, health, and clothing. Evidence shows that mothers in disadvantaged houses serve food first to their husbands and sons and then to their daughters. Deficient and poor-quality food is left for the daughters [11]. Due to this, poverty females are victimized not only in nutritional intake and health outcomes but also in other social and economic terms [12].

In Pakistan, numerous studies have observed the determinants of child malnutrition. The most important variables of child malnutrition in the literature over time are a poor economic status of a household and maternal education status. Recent literature in Pakistan highlighted that richer households have a lower probability of stunting and wasting prevalence [13]. Another case study from the deprived region of Punjab in Pakistan revealed that malnutrition chances decrease with an increase in the wealth status from rich to richest [14]. The results of another study from the rural areas of Punjab depicted that children belonging to richer families have less distance to travel to healthcare facilities and regular visits from an LHW within 15 days reduces the chances of malnutrition [15]. Another study from the rural areas of Punjab revealed that as the household shifts from a lower socio-economic status (HDS-1) to a middle socio-economic status (HDS-2) and further to a rich socio-economic status (HDS-3), the rates of being underweight and stunting decreases [16]. The educational status of the mother is also a leading determinant of the child malnutrition status in Pakistan. In the literature, it has been observed that the education of mothers has a direct and indirect link with the child malnutrition status. The direct effect shows the understanding of child healthcare and breastfeeding practices of mothers [17,18,19]. The indirect effect of maternal education works through late marriages of women, reducing the demand of the children and the empowerment of women [20,21,22].

Several research findings have revealed several crucial determinants of child malnutrition, including water quality and sanitation facilities. Water, sanitation, and hygiene (WASH) belong to the four “pillars” of the Food and Nutrition Protection Framework [23]. A few studies regarding child malnutrition have discovered that WASH is considered to be a prominent determinant in South Asia [24,25,26], including in Pakistan [27,28]. The literature shows that inadequate situations for WASH in developing countries are the main reason children miss school and the odds of sexual abuse and the rape of girls increase [29,30,31]. Other studies have revealed that households having access to sanitation and clean water reduced the danger of stunting [32,33,34] and child mortalities [35,36,37].

Studies have shown that infants and children who are more at risk of stunting experience diarrhea and gastrointestinal contagions, which are correlated with bad WASH situations and open defecation [38,39]. The UNICEF and WHO supported WASH in developing countries as a prime approach for fighting child malnutrition, diarrhea, and morbidities [40]. The evidence on nutritional discrimination in children in Pakistan is limited. Moreover, the gender-wise impact of sanitation facilities and water sources on malnutrition has never been systematically assessed in Pakistan. Therefore, to bridge this gap, the current study attempted to examine how water and sanitation indicators behaved in a gender-disaggregated analysis in Pakistan.

## 2. Methods

### 2.1. Data

This study used the 2018 PDHS data, with a sample of 1010 children under the age of three. Out of 1010, 497 (49.21%) were female and 513 (50.79%) were male children under three. The data gave broad information on the empowerment of women, the healthcare and nutritional information of women and children, nutrition and demographic characteristics, and domestic violence, for example.

### 2.2. Dependent Variables

In the present study, a CIAF was established to quantify child nutrition based on three indices (weight-for-age (WAZ), weight-for-height (WHZ), and height-for-age (HAZ)), conferring to the WHO (2009) standards of growth [41]. The CIAF classification had seven different categories for the children: 1: underweight only; 2: stunted only; 3: wasted only; 4: no failure; 5: both underweight and wasting both; 6: both underweight and stunted; and 7: wasted, underweight, and stunted. The scale of malnutrition prevalence in children was entirely measured by the sum of all, excluding group A. For a child with no malnutrition, the binary variable “0” was used and “1” was used for a malnourished child.

### 2.3. Independent Variables

This study was based on the conceptual framework of Victoria et al. [42], who proposed a conceptual framework based on the previous literature for studying and predicting the determinants of the health outcomes of children [42,43]. According to them, the distribution of variables in this framework is in three groups: (1) socio-economic indicators (e.g., region/place of residence, educational level of the mother, maternal employment status, BMI of the mother, age of the mother at the birth of their child, and household wealth status); (2) intermediate factors, including environmental determinants (e.g., the type of latrine/sanitation, the source of water, and the type of house); and (3) proximal or individual factors (e.g., the child birth order, sex of the child, gender of the child, weight-for-age, weight-for-height, height-for-age, and child diseases). In short, the nutritional status of preschool children may be affected by these factors [42,43].

The main explanatory variables in this study were the drinking water source and the form of sanitation/toilet facilities. To compute the two variables, the responses on the water source and sanitation facility were separated into categories, which were unimproved sanitation, improved sanitation, unimproved water, and improved water. These variables were binary in the analysis. In the analysis, we assigned the value “1” if the source of water was improved and “0” if unimproved. In the same way, “1” was assigned if the type of sanitation facility was improved and “0” if unimproved. The composition of the responses is given in Table 1.

The other socio-economic factors included in the analysis as possible determinants or control variables were the age of the child in months, sex of the child, education level of the mother, region, had diarrhea recently, and wealth index. The conceptual relationship between the explanatory variables of the study or the socio-economic and environmental factors with the child malnutrition status, based on the conceptual framework of Victoria et al. [43], is shown in Figure 1:

### 2.4. Statistical Analysis

The study measured the relationship between the socio-economic variables and malnourishment through the use of a logistical regression. It estimated the odds ratio and probability score of the resultant variables (CIAF), subject to the explanatory variables.

The binary logistic model had the following explanation:
CIAFin = Yin= (1 = for malnourished child, and 0 = otherwise)
where Y in the equation shows the malnutrition status of a child. The study evaluated whether the other various explanatory variables (X1, X2… Xn) were associated with it. The response was twofold; either failure or success as to whether the child was malnourished or not with a code of “0” or “1”. The sample weights were applied to all results.

## 3. Results

This study explored the effect of sanitation/toilet facilities and water sources on children under three with malnourishment. Analyzing the data (PDHS) for all variables, the percentage of malnutrition (CIAF) prevalence in children concerning various features was obtained and is given in Table 2, depicting that the malnourishment prevalence among children below three years of age was 252 (50.50%) in males and 247 (49.50%) in females. Out of 1010 children, 499 (49.41%) were malnourished and 511 (50.59%) were normal or not malnourished. A total of 245 (47.95%) male and 266 (52.05%) female children were not malnourished; in other words, they were normal, so the percentages and frequencies of the normal children are not listed in Table 2. The malnutrition rates were higher in children aged between 19 and 36 months. The prevalence of malnutrition was high in the Sindh province (24.65%), followed by KPK (15.43%) and Balochistan (24.85%). Around 73.15% of the malnourished children were from households where the mothers were illiterate. The prevalence of malnutrition among the children under three was 37.88% in the poorest households and 29.26% in poorer households. Of the malnourished children, 26.25% had recently had diarrhea. The malnutrition prevalence was 26.45% where unimproved water sources were used and 36.27% where unimproved sanitation facilities were used.

### Malnutrition Association with the Water Source and Sanitation Facilities

Figure 2a,b highlight the rates of prevalence of being underweight, stunting, and wasting by the gender of the children. The results of both figures display that the rate of stunting and being underweight showed a reduction as households enhanced their sanitation facilities whereas the rates of wasting due to a change in sanitation facilities remained constant.

Figure 3a,b illustrate the rates of prevalence of being underweight, stunting, and wasting by the gender of the children. The results of both figures display that the being underweight and stunting rates declined as households enhanced their sanitation facilities. The rates of wasting also decreased in boys as households enhanced their water sources, but the rates of wasting did not change (remaining constant) in girls.

In Table 3, we report the results of the logistic regression. Model I estimated the coefficients of the CIAF for the male children and model II estimated the coefficients of the CIAF for the female children.

Children aged 19–36 months were more likely to be malnourished in both cases. The malnutrition odds were higher in the male children belonging to the Sindh (OR = 0.80, CI = 0.29–2.24) and Balochistan (OR = 3.03, CI = 1.5–6.33) regions whereas the female children in the KPK (OR = 2.07, CI = 0.99–4.29), Sindh (OR = 3.15, CI = 1.5–6.45), and Balochistan (OR = 3.44, CI = 1.6–7.47) regions/provinces were bound to be malnourished. A primary school education of the mothers reduced the chances of malnutrition in the male children (OR = 0.49, CI = 0.25–0.95); in female children, a secondary education of the mothers was related to fewer chances of malnourishment among their children (OR = 0.53, CI = 0.27–1.04). In terms of wealth, children from richer households were less likely to suffer from malnutrition (OR = 0.48, CI = 0.22–1.02) as well as the richest wealth index quantiles (OR = 0.33, CI = 0.13–0.81); malnutrition was also the least in the richest WIQ in female children (OR = 0.48, CI = 0.20–1.13). Malnutrition was thought to be more likely in male children who had recently had exposure to diarrhea (OR = 1.55, CI = 0.96–2.50).

The results of the source of drinking water in the male model showed a lesser probability of male child malnutrition where there were improved water sources in their houses (OR = 0.62, CI = 0.37–1.03). The sanitation variable was not significant in the male model. Similarly, the results of the sanitation facilities in the female model illustrated that households with enhanced sanitation facilities had lower chances of female malnutrition (OR = 0.64, CI = 0.43–0.95). The water source variable was not significant in the female model.

## 4. Discussion

This study analyzed the impact of sanitation facilities and drinking sources of water on a gender analysis for the determinants of child malnutrition in Pakistan using nationally representative data from PDHS 2017–18. A sample of 1010 children under three was included in the analysis. Around 36.37% of malnourished children belonged to those households that used unimproved sanitation facilities and 26.45% of malnourished children belonged to those households that had unimproved water sources. Around 67.14% of malnourished children under three belonged to the poorest and poorer households in Pakistan (37.88% in the poorest and 29.26% in the poorer). 

Overall, the results of logistic regression analysis in the gender analysis for the determinants of child malnutrition in children under three are presented in Table 3. The logistic regression results of the study showed that children aged 19–36 months were more likely to be malnourished in both cases. It portrayed that with an increase in age, the prevalence of malnutrition also increased. The results were consistent with the previous literature [44,45,46]. The consequences of the study were that the chances of malnutrition were higher in children belonging to the Sindh, Balochistan, and KPK regions/provinces. There is a regional diversity in nutrition in Pakistan. Compared with Punjab and KPK, women in Sindh and Balochistan have higher levels of nutrition in adulthood or at a younger age [47]. Another study showed that the prevailing modes of malnutrition in Balochistan and Sindh were greater than in other parts of Pakistan as these provinces are deprived in terms of socio-economic development [3]. The study results depicted that the primary and secondary education of the mother lessened the malnutrition odds in both female and male children. The findings of the study were also in accordance with previous studies that observed that the significant determinant of malnutrition is the education of the mothers [48,49,50,51]. The results of this study demonstrated that the odds of both male and female malnutrition were low in the rich and richest wealth status households. Previous studies have depicted that children have a lower probability of being malnourished as their parents have more resources to purchase food items [52,53,54]. Diarrhea, respiratory infections, and malnutrition are three major causes of death during infancy [55]. The results of our study illustrated that the malnutrition odds were high among only male children who had recently had diarrhea. Studies have shown that the prevalence of diarrhea was higher in male children compared with their female counterparts; this had a significant effect on male child malnutrition [56,57].

The results illustrated that households with improved drinking water sources had lower chances of malnutrition only in male children whereas an improved sanitation facility had lower chances of malnutrition only in female children. According to a study conducted in Pakistan, sanitation and water are closely related to malnutrition in Pakistan and there is little improvement in sanitation and water in the SAARC regions [58]. Previous studies have depicted that male children are at a higher risk of contracting schistosomiasis (water-borne disease) and stunting because of unimproved water [32,59,60]. In Nigeria, boys were seven times more likely to be infected with schistosomiasis and stunting compared with girls [59]. Research in Ghana and West Africa revealed that enhanced hygiene and water reduced the stunting risk by 15% in boys compared with girls [61]. Admittance to enhanced drinking water decreased the probability of stunting by 30% in male children compared with their female counterparts in Nigeria [32]. A study in India showed that poor access to sanitation facilities and a low infant birth weight were significantly associated with female harassment [62]. A study concluded that unimproved sanitation facilities were highly linked with maternal mortality [63]. In Kenya, a study depicted that those households in which the head was female had more chances to openly defecate compared with their male counterparts [64]. In Ethiopia, mothers were more likely to be underweight and have an underweight child who was fasting and with no access to enhanced sanitation [65]. Another study in Ethiopia showed that poor sanitation facilities and poor water sources lowered the probability of wasting [66]. A study conducted in Pakistan showed that sanitation and water were the only important components of the interaction of the variables (sanitation and water), indicating that homes with improved water sources and sanitation facilities had fewer chances of undernutrition in their children [67].

### Strengths and Limitations of the Study

The strength of this study was its novel analysis of the gender-wise effects of water and sanitation/toilet facilities on the malnourishment of children under three in Pakistan. Under three is a vital age for child growth. At this age, children are more prone to diseases and infections than above the age of three and they need more care. Observing the determinants for children under three is vital for policy decisions. Using a national dataset, the results and scope of this study can be generalized for policy solutions. However, this study has a few limitations as it used secondary data with a prearranged set of variables to meet our objective, which meant that other important variables (i.e., dietary diversity, inadequate access to food, and human and organizational resources as well as political, cultural, and ideological factors) were either missing or not included in our analysis because of their insignificance in the study model.

## 5. Conclusions

The results of the study illustrated that an improved water source significantly reduced the probability of malnutrition in male children whereas an improved sanitation facility lowered the probability of malnutrition in the case of the female child in Pakistan. The education of the mother, a child belonging to deprived regions, and the household wealth status were the significant determinants of both male and female child malnutrition. Therefore, improved facilities of sanitation and water are highly recommended to achieve the targets in sustainable development goal (SDG) 6. This will also help to meet the other goals of sustainable development such as SDG 3 (wellbeing and good health) and SDG 4 (gender equity). The study concludes that improved sanitation and water facilities are direly needed for human health and wellbeing in Pakistan.

## Figures and Tables

**Figure 1 children-09-01674-f001:**
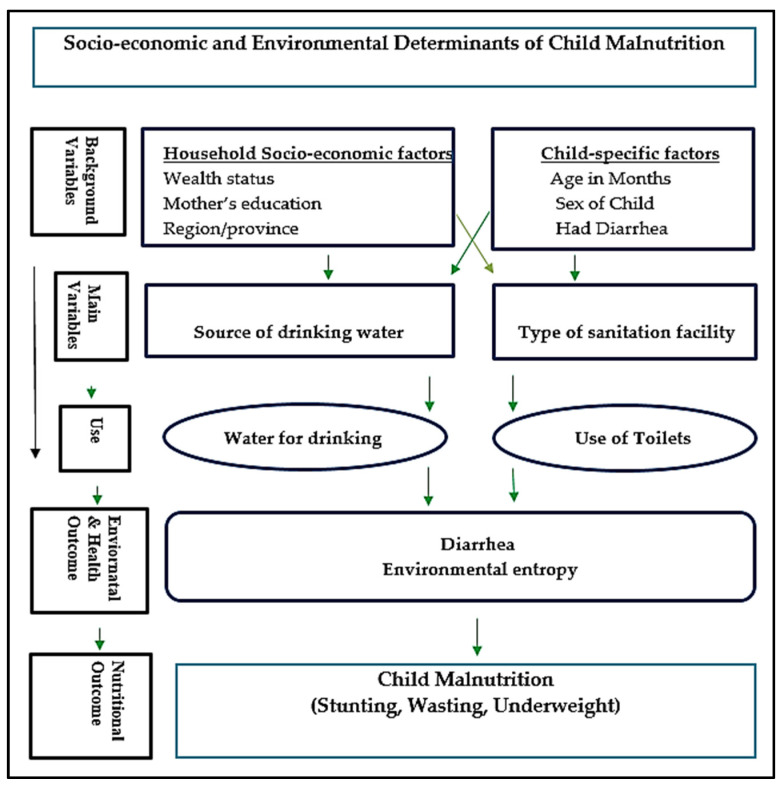
The conceptual relationship between the explanatory variables of the study and child growth status.

**Figure 2 children-09-01674-f002:**
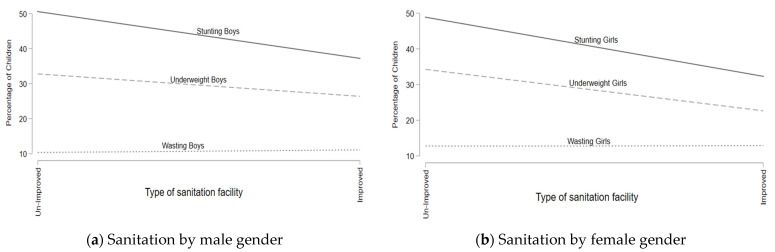
Prevalence rates of wasting, stunting, and being underweight for sanitation facility (disaggregated by child gender).

**Figure 3 children-09-01674-f003:**
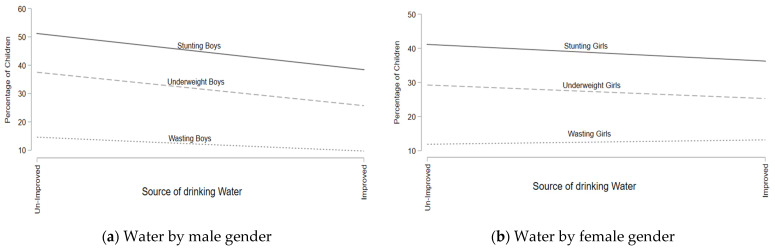
Prevalence rates of stunting, wasting, and being underweight for water source (disaggregated by child gender).

**Table 1 children-09-01674-t001:** Description of water and sanitation variables.

Drinking Water Source		Sanitation/Toilet Facility Type	
Improved	Unimproved	Improved	Unimproved
Piped water Piped into lodging Piped water into plot/yard Public tap/standpipe Borehole Tube well water Protected spring Rainwater Protected well Filtration plant	Piped to neighbor Unprotected dug well Protected well Tanker truck Unprotected spring Bottled/sachet water Surface water River/stream/lake/dam/canals/pond Cart with small tank Other unimproved	Flush latrine Flush to pit latrine Flush/pour flush to piped sewer system Flush to septic tank Pit toilet latrine Pit latrine with slab Ventilated improved pit latrine (VIP) Compositing toilet	Pit latrine without a slab or open ditch Bucket toilet Any facility shared with other households Hanging toilet/latrine Flush to somewhere else No facility Facility/stream/field/river/bush Flush, do not know where Other unimproved

**Table 2 children-09-01674-t002:** Prevalence of malnutrition by different socio-economic factors.

Variables	Categories	*F*	%
Sex of Child	Male	252	50.50
	Female	247	49.50
Age of Child (Months)	0 to 6	69	13.83
	7 to 12	60	12.02
	13 to18	62	12.42
	19 to 24	81	16.23
	25 to 36	227	45.49
Region	Punjab	49	9.82
	Sindh	123	24.65
	KPK	77	15.43
	Balochistan	124	24.85
	Gilgit Baltistan	23	4.61
	ICT (Capital)	14	2.81
	AJK	26	5.21
	FATA	63	12.63
Qualification Level of Mother	Illiterate	365	73.15
	Primary	48	9.62
	Middle	59	11.82
	High	27	5.41
	11–15	130	26.05
	Greater than 15	73	14.63
Wealth Index	Poorest	189	37.88
	Poorer	146	29.26
	Middle	74	14.83
	Richer	52	10.42
	Richest	38	7.62
Source of Drinking Water	Unimproved	132	26.45
	Improved	367	73.55
Had Diarrhea Recently	No	368	73.75
	Yes	131	26.25
Sanitation Facility Type	Unimproved	181	36.27
	Improved	318	63.73

Source: estimation of the authors.

**Table 3 children-09-01674-t003:** Logistic regression showing adjusted odd ratios for the covariates of the CIAF.

		Model I	Model II
Variables	Categories	OR (95% CI)	OR (95% CI)
Age of Child (Months)	0 to 6 months (R)		
	7–12 months	1.15 (0.56, 2.36)	1.13 (0.56, 2.25)
	13–18 months	0.80 (0.39, 1.63)	1.43 (0.72, 2.84)
	19–24 months	2.10 ** (1.00, 4.4)	2.57 *** (1.28, 5.2)
	25–36 months	1.93 ** (1.00, 4.4)	3.87 *** (2.2, 6.78)
Region	Punjab (R)		
	Sindh	2.89 *** (1.4, 5.85)	3.15 *** (1.5, 6.45)
	KPK	1.66 (0.79, 3.47)	2.07 ** (0.99, 4.29)
	Balochistan	3.03 *** (1.5, 6.33)	3.44 *** (1.6, 7.47)
	Gilgit Baltistan	0.80 (0.29, 2.24)	0.97 (0.37, 2.55)
	ICT (Capital)	1.59 (0.56, 4.58)	0.58 (0.16, 2.07)
	AJK	1.28 (0.44, 3.75)	1.16 (0.49, 2.77)
	FATA	1.26 (0.54, 2.91)	1.22 (0.55, 2.72)
Qualification Level of Mother	Illiterate (R)		
	Primary	0.49 ** (0.25, 0.95)	0.88 (0.45, 1.71)
	Middle	1.03 (0.53, 1.98)	0.53 ** (0.27, 1.04)
	High	0.35 (0.15, 0.80)	0.51 (0.23, 1.17)
Wealth Index	Poorest (R)		
	Poorer	0.92 (0.52, 1.65)	0.83 (0.48, 1.46)
	Middle	0.60 (0.29, 1.26)	0.92 (0.46, 1.85)
	Richer	0.48 ** (0.22, 1.02)	0.65 (0.29, 1.47)
	Richest	0.33 *** (0.13, 0.81)	0.48 * (0.20, 1.13)
Had Diarrhea Recently	No (R)		
	Yes	1.55 * (0.96, 2.50)	1.48 (0.93, 2.36)
Drinking Water Source	Unimproved (R)		
	Improved	0.62 ** (0.37, 1.03)	0.80 (0.49, 1.31)
Sanitation Facility	Unimproved (R)		
	Improved	1.24 (0.75, 2.03)	0.64 ** (0.43, 0.95)
Overall Significance of the Model			
Number of Observations		497	513
LR Chi^2^		90.42	96.57
Prob > Chi^2^		0.0000	0.0000
Pseudo R^2^		0.1313	0.1359

Source: estimation of the authors. References: odds ratios; confidence intervals. R: reference category in the model. Significance level: *** if *p* < 1%; ** if *p* < 5%; * if *p* < 10%.

## Data Availability

Data from the Pakistan Demographic and Health Survey (2017–2018) were utilized in the study (available online at: https://dhsprogram.com/data/dataset/Pakistan_Standard-DHS_2017.cfm?flag=1 (accessed on 20 June 2020).

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
