# Peer review of "Impact of Drinking Water Source and Sanitation Facility on Malnutrition Prevalence in Children under Three: A Gender-Disaggregated Analysis Using PDHS 2017–18"

_children, 2022, doi:10.3390/children9111674_

Round 1
Reviewer 1 Report
The article analyzes important aspects for improving the quality of life of children whose lives are affected by socio-economic factors.
My recommendations are the following:
Abstract: reformulation of objectives.
Introduction: Reformulation of information because some ideas are repeated (lines 37-42 and 46-52)
- more information can be provided related to the influence of socio-economic status in relation to the other variables.
Material and methods:
- what kind of study is it (see https://www.strobe-statement.org/)
- the inclusion and exclusion criteria of the subjects.... maybe that's how we found out how we got from 1010 children to 499 children
- explain the abbreviations in the text.
In the attached article you will find all my observations.

Author Response
Comment 1: Abstract: reformulation of objectives.
Response by Author: Thanks for your valuable comment. We have revised our study objective in the abstract section of the revised draft. The proposed research studies the determinants of male and female child malnutrition in Pakistan. More specifically, it observes the role of the sanitation facility and drinking water sources as important determinants of malnutrition in gender analysis.
(Kindly see lines 19-21)
Comment 2: Introduction: Reformulation of information because some ideas are repeated (lines 37-42 and 46-52) - more information can be provided related to the influence of socio-economic status in relation to the other variables.
Response by Author: Thanks for your valuable comments. We have revised the introduction section according to your suggestion. But along with that, we want to elucidate that the information from 37 to 52 was under the umbrella of gender-wise discrimination in nutrition and socioeconomic division because this study is based on gender analysis for the determinants of malnutrition. We think that we should give literature in the context of gender before going into gender analysis. Moreover, the sex and age of the child are also major determinants of malnutrition as we also explained the age and sex of the child in this regard. It does not mean that we are going to attempt new themes on gender discrimination in food diversity or nutritional intake. So, we decided to keep that part in the paper. In addition, we added literature in the context of socioeconomics or wealth status as per your suggestions. Now improvement in the introduction is as follows:
This study is focused on gender-based determinants of child malnutrition, more focused on water and sanitation as an important factors of under three child malnutrition in Pakistan. In developing countries, the most common phenomenon is sex bias or gender preferences, or discrimination in poor households. The bad principles found in poor families include better care for their sons than for their female children. [10]. Preference for sons in poor societies means sons will be the source of income for the family but females are a burden [10]. This preference belief of parents also creates discrimination every time in food, education, health, and clothing. Injustice in food distribution is also there in disadvantaged houses in which mothers first feed their husbands and sons and then offer it to their daughters and often poor quality and deficient food is left for the daughters [11]. Poverty in rural as well as poor urban areas is higher in Pakistan, and girls and women are socioeconomically victimized more than in nutritional intake and health outcomes [12].
In Pakistan, numerous studies have observed the determinants of child malnutrition. The most important variable of child malnutrition in literature over time are household poor economic status and maternal education status. Recent literature in Pakistan highlighted that richer households have a lower probability of stunting and wasting prevalence [13]. Another case study from the deprived region of Punjab Pakistan reveals that malnutrition chances decrease with the increase in wealth status from rich to richest [14]. Results of another study from rural areas of Punjab, Pakistan depicted that child belongs to richer families, have less distance to healthcare facilities, and LHW regular visits within 15 days reduces the chances of malnutrition [15]. Another study from rural areas of Punjab Pakistan reveals that as the household shifts from lower socio-economic status (HDS-1) to middle socio-economic status (HDS-2) and further to rich socio-economic status (HDS-3), the rates of underweight and stunting decreases [16]. Mother's educational status was also a leading determinant of child malnutrition status in Pakistan. In literature, it is observed that the education of mothers has a direct and indirect link with child malnutrition status. The direct effect shows mothers' understanding of child healthcare and breastfeeding practices [17-19]. An indirect effect of maternal education works through women's late marriages which reduces the children's demand, and women's empowerment [20-22].
Several research findings reveal some crucial determinants of child malnutrition which include water quality and sanitation facilities. Water, Sanitation, and Hygiene (WASH) belong to the four "pillars" of the Food and Nutrition Protection Framework [23]. A few studies regarding child malnutrition discover that WASH is considered a prominent determinant in South Asia [24-26], including in Pakistan [27-28]. Literature shows inadequate situations for WASH in developing countries is the main reason children miss school and the odds of sexual abuse and girls’ rape increase [29-31]. Other studies reveal that households having access to sanitation and clean water reduces the danger of stunting [32-34], and child mortalities [35-37].
(Kindly see lines 45-83)
Comment 3: Material and methods: - what kind of study is it - the inclusion and exclusion criteria of the subjects.... maybe that's how we found out how we got from 1010 children to 499 children.
Response by Author: Thanks for your valuable comment. This study used secondary data set from Pakistan Demographic and Health Survey which is publicly available at: https://dhsprogram.com/data/dataset/Pakistan_Standard-DHS_2017.cfm?flag=1. The inclusion criteria of this study are under three children. Under three age is a vital age for child growth. At this age, children are more prone to diseases and infections than above three age and they need more care. Observing the determinants for under-age three children would be vital for policy decisions. (Kindly see lines 270-273 in the limitation of this study section)
While the total sample is 1010 under three children, out of them 513 female children and 497 male children. Regression analysis is also on the 1010 sample. While in Table 2 we have mentioned the results of crosstabulations of malnutrition (CIAF) against different variables. Results are accurately presented in Table 2. Because we only showed malnourished children in Table 2 and ignore to present normal/not-malnourished categories results.
The percentage of Malnutrition (CIAF) prevalence in a child concerning various features is given in Table 2, which depicted that malnourishment prevalence among below three years of age children was 252 (50.50%) in males and 247 (49.50%) in females. Out of 1010 children, 499 (49.41%) children were malnourished and 511 (50.59%) children were normal or not malnourished. While 245 (47.95%) male and 266 (52.05%) female children were not malnourished in other words they were normal, so we did not mention percentages and frequencies of normal children in Table 2.
(Kindly see lines 152-158)
Cross tabulation for malnutrition (CIAF) by Gender
|
|
Not-malnourished |
Malnourished |
Total |
|
Male |
245 (47.95%) |
252 (50.50%) |
497 (49.21%) |
|
Female |
266 (52.05%) |
247 (49.50%) |
513 (50.79%) |
|
Total |
511 (50.59%) |
499 (49.41%) |
1010 (100%) |
Comment 4: explain the abbreviations in the text. In the attached article you will find all my observations.
Response by Author: Thanks for your valuable comment. We have added the full information against abbreviations in the revised draft which are:
Stunting= height-for-age (HAZ); Underweight= weight-for-age (WAZ); wasting= weight-for-height (WHZ). Moreover, we also added an abbreviation for SDGs in the conclusion (sustainable development goals).
(Kindly see line 102)
Comment 5: Modification/correction in references
Response by Author: Thanks for your correction.
- Comment on reference 3: we checked is perfect on MDPI website also mentioned on paper citation after downloading the PDF file of paper from MDPI. It is 4566 not 45-46.
- Comment on reference 7: we revised the link after check and now this reference is accessible at https://www.emerald.com/insight/content/doi/10.1108/JHASS-02-2021-0030/full/html
- Comment of reference 8: we have corrected this after rechecking again which is 47-59.
- Comment on reference 18: corrected as per suggestion which is 103-109.
- Comment of reference 19: we have checked and revised this link again now accessible at: https://gbvguidelines.org/wp/wp-content/uploads/2015/09/2015-IASC-Gender-based-Violence-Guidelines_lo-res.pdf
While as an alternative: https://interagencystandingcommittee.org/working-group/iasc-guidelines-integrating-gender-based-violence-interventions-humanitarian-action-2015
Reviewer 2 Report
Dear Author,
thank you for your interesting manuscript, however it needs high level of improvement in term of methodology, result and discussion presentation.
Th writing is not clear, results are presented poorly and understandable.
Author Response
Comment/complement by reviewer: Thank you for your interesting manuscript, however it needs high level of improvement in term of methodology, result and discussion presentation. Th writing is not clear, results are presented poorly and understandable.
Response by Author: Thanks for your comment. We have revised our entire paper along with the suggested comments by other reviewers. Now the paper is according to your expectations.
Reviewer 3 Report
Interesting and important topic, though more work needs to be done on the manuscript.
1. It is unclear if you controlled for dietary intake, dietary diversity, or food security status. This might be confounding the results if you did not. Are your results truly the result of inadequate WASH? Or is the malnutrition in these households due to the fact that they also have insufficient food? If you controlled for this, make it clear in your methods. If not, I recommend doing so.
2. Similarly, your conceptual framework talks about sociodemographic and household factors but ignores food security and several other socioeconomic and environmental determinants. Also, in your introduction you mention gender issues but these are not in the conceptual framework. Did you develop the framework yourselves? If not, the source is not included? Please refer to the UNICEF Conceptual Framework of Malnutrition for guidance.
3. DHS recommends that their data should be weighted when analyzing their datasets (https://dhsprogram.com/data/Guide-to-DHS-Statistics/Analyzing_DHS_Data.htm). It is unclear if you did this.
4. English editing is needed as there are a number of errors and sections that need to be rephrased.
5. I do not think the sentence with citation 53 is relevant to the study.
Author Response
Comment 1: It is unclear if you controlled for dietary intake, dietary diversity, or food security status. This might be confounding the results if you did not. Are your results truly the result of inadequate WASH? Or is the malnutrition in these households due to the fact that they also have insufficient food? If you controlled for this, make it clear in your methods. If not, I recommend doing so.
Response by Author: Thanks for your valuable comment. First of all, we would like to explain that this study is all about determinants of malnutrition focusing on water and sanitation as main explanatory variables. The point you are mentioning is to incorporate such as dietary intake, dietary diversity, or food security status are the theme for purely food security-related issues, and nutrition or calorie intake approaches, something different to deal with separately in detail. However, we included it in our study limitations:
“However, this study has some limitations as it used secondary data with a prearranged set of variables to meet our objective, which means the other important variables i.e., dietary diversity, inadequate access to food, human and organizational resources, political, cultural, and ideological factors were either missing or either not included in our analysis because of their insignificance in study model.” (See the line from 274-279)
But we also again attempted our analysis just to verify. In Pakistan Demographic and Health Survey there are indicators related to dietary intake. We included few variables which were in binary form (Yes or No) included in model which are: 1) give child powered or fresh milk [odds=1.05; p-value=0.82]; 2) give child baby formula milk [odds=1.64; p-value=0.2]; 3) give child fortified baby food such as cerelac etc. [odds=0.64; p-value=0.2]; 4) give child food made from beans, peas, lentils, nuts etc. [odds=0.65; p-value=0.22]; 5) give child vitamin A supplements [odds=4.74; p-value=0.09*]. Out of these 5 variables, 4 were insignificant, while only 1 variable vitamin A supplements were significant but it shows that giving vitamin A supplements to child, higher the chances of malnutrition which makes no sense. So, we decided not to add this section in our paper at this stage and to justify for we need the entire discussion to change. We achieved our objective with the combination of provided variables, and theme was very simple based on the conceptual framework of Victoria et al (1997). So, in other words, you can say results are purely for WASH representation.
But there are a few reasons not to include dietary intake in our paper:
- Results were insignificant in the second analysis after your suggestion for dietary and food-related indicators in our model.
- There are two approaches to measuring the nutritional status of children. 1) food intake or calorie intake approach – which is based on the nutrition intake approach; 2) health outcome or anthropometric approach – which is based on physical measurements. We adopted an anthropometric approach.
- We think food or calorie intake-related indicators are related to the first nutritional assessment approach so no need to visualize or handle food intake-related variables.
Comment 2: Similarly, your conceptual framework talks about sociodemographic and household factors but ignores food security and several other socioeconomic and environmental determinants. Also, in your introduction you mention gender issues but these are not in the conceptual framework. Did you develop the framework yourselves? If not, the source is not included? Please refer to the UNICEF Conceptual Framework of Malnutrition for guidance.
Response by Author: Thanks for your valuable comment. First of all, this study is all about determinants of child malnutrition by gender. Within the study’s objective, special attention was given to water and sanitation as the main explanatory variable in gender analysis. Moreover, The components in UNICEF conceptual framework are different as compared to ours i.e., they talk about dietary intake, resources, and controls (human resources, organizational resources), insufficient health services, inadequate access to food, political and ideological factors, etc. These components are not part of our study because of two reasons: 1) the first limitation with the data set as we have used secondary cross-sectional data set; 2) this study followed the conceptual framework of Victoria et al (1997).
This study is based on the conceptual framework of Victoria et al (1997). They have proposed a conceptual framework based on previous literature for studying and predicting the determinants of children's health outcomes [42-43]. According to them, the distribution of variables in this framework is in three groups: 1) socio-economic indicators (region/place of residence, educational level of mother, maternal employment status, mother BMI, mother age at birth, household wealth status, etc.), 2) intermediate factors include environmental determinants (type of latrine/sanitation, source of water type of house, etc.), and 3) proximal or individual factors (child birth order, sex of the child, gender of the child, weight-for-age, weight-for-height, height-for-age, child diseases, etc.). In short, preschool children’s nutritional status may be affected by these factors [42-43].
(See the line from 110-119)
Moreover, we revised the title of Figure-1 which is: Conceptual relationship between study’s explanatory variables and child growth status [based on the conceptual framework of Victoria et al (1997)]. The conceptual relationship between the study’s explanatory variables or socioeconomic and environmental factors with child malnutrition status based on the conceptual framework of Victoria et al (1997) was shown in Figure 1:
(See lines 132-138)
As for your concern on gender is concerned: we want to explain that the gender information was under the umbrella of gender-wise discrimination and gender-wise socio-economic division because this study is based on gender analysis for the determinants of malnutrition that’s why we think that we should support literature in the context of gender before going into gender analysis. Moreover, the sex and age of the child are also major determinants of malnutrition as we also explained the age and sex of the child in this regard. It does not mean that we are going to attempt new themes on gender discrimination in food diversity or nutritional intake. Overall, in Pakistan, gender base work is scarce, especially in terms of how water and sanitation variable behaves in gender analysis. This paper is on the track to provide important baseline literature on gender-based water and sanitation analysis under the umbrella of determinants of child malnutrition.
Comment 3: DHS recommends that their data should be weighted when analyzing their datasets (https://dhsprogram.com/data/Guide-to-DHS-Statistics/Analyzing_DHS_Data.htm). It is unclear if you did this.
Response by Author: Thanks for your valuable comment. We have already applied sample weights in our initial draft. For more clarification, we have mentioned it in 2.4. statistical analysis section in our revised draft now.
(See line 148)
Comment 4: English editing is needed as there are a number of errors and sections that need to be rephrased.
Response by Author: Thanks for your valuable suggestion. We have checked errors, and corrected and revised our draft according to your suggestion.
Comment 5: I do not think the sentence with citation 53 is relevant to the study.
Response by Author: Thanks for your valuable comment. As per your suggestion, we excluded citation 53 in our revised draft.
Round 2
Reviewer 2 Report
This manuscript reports findings from a longitudinal study investigating
associations between impact of Drinking Water Source and Sanitation Facility on
Malnutrition Prevalence in Under Three Children: A Gender-Disaggregated Analysis
Using PDHS 2017-18. Authors gave insight into a very important topic.
The manuscript is well organized and corrected, however it needs extensive editing of English language and style.
Author Response
Comment/complement by reviewer: The manuscript is well organized and corrected, however it needs extensive editing of English language and style.
Response by Author: Thanks for your comment. We have revised our entire paper. Now editing and style of the paper are according to your expectations and journal requirement.
Reviewer 3 Report
- Most of the methodology-related concerns I had raised have been addressed.
- Line 75 “In developing countries, the most common phenomenon is sex bias or gender preferences, or discrimination in poor households.”- Is this statement true? Is sex bias or gender preference the “most common phenomenon in developing countries”? If so, a citation is needed.
- The document still requires language editing. Several statements need to be rephrased for clarity. For example:
- These are just 2 examples, but there are several other sections throughout the text that require language editing.
Author Response
Comment 1: Line 75 “In developing countries, the most common phenomenon is sex bias or gender preferences, or discrimination in poor households.”- Is this statement true? Is sex bias or gender preference the “most common phenomenon in developing countries”? If so, a citation is needed.
Response by Author: Thanks for your valuable comment. We have again revised our statement more clearly for batter understanding. Citation 8 and 10 explain this statement. Now after phrasing literature is given in introduction with proper citations below:
Studies show that male children are more preferred by their parents and socio-economically girls are more deprived than boys. Gender discrimination at home makes girls more malnourished than male children. [4-6]. However, some researchers argued that male children are at higher levels of malnourishment compared to their female counterparts due to the poor socioeconomic status of the household [7-9]. Gender, as well as nutrition, is a tricky fragment of a poverty vicious cycle [8]. In developing countries, the most common phenomenon is sex bias or gender preferences, or discrimination in poor households [10]. Unfortunately, in many poor families sons are preferred over their daughters [10]. The main reason is that males are considered breadwinners while females are perceived as a burden on the poor family [10]. This preference compels many poor parents to discriminate in food, education, health, and clothing. Evidence shows that mothers in disadvantaged houses first serve food to their husbands and sons and then to their daughters. Also, deficient and poor-quality of food is left for the daughters [11]. Because of this poverty females are victimized not only in nutritional intake and health outcomes but also in other social and economical terms [12].
(See lines 38-56)
Comment 2: The document still requires language editing. Several statements need to be rephrased for clarity. For example: Rephrase line 43 “Gender, as well as nutrition, is a tricky fragment of a poverty vicious cycle.”
Response by Author: Thanks for your valuable comment. We have further improved our editing. We have rephrased line 43-45 which is given below:
Undernourished girls in their adult life produce malnourished babies and a vicious cycle of undernutrition and poverty continues [8].
Comment 3: Rephrase line 76-84 “The bad principles found in poor families include better care for their sons than for their female children [10]. Preference for sons in poor societies means sons will be the source of income for the family but females are a burden [10]. This preference belief of parents also creates discrimination every time in food, education, health, and clothing. Injustice in food distribution is also there in disadvantaged houses in which mothers first feed their husbands and sons and then offer it to their daughters and often poor quality and deficient food is left for the daughters [11]. Poverty in rural as well as poor urban areas is higher in Pakistan, and girls and women are socioeconomically victimized more than in nutritional intake and health outcomes [12].
Response by Author: Thanks for your valuable suggestion. We have rephrased our lines 76-84 which are as follows:
Unfortunately, in many poor families sons are preferred over their daughters [10]. The main reason is that males are considered breadwinners while females are perceived as a burden on the poor family [10]. This preference compels many poor parents to discriminate in food, education, health, and clothing. Evidence shows that mothers in disadvantaged houses first serve food to their husbands and sons and then to their daughters. Also, deficient and poor-quality of food is left for the daughters [11]. Because of this poverty females are victimized not only in nutritional intake and health outcomes but also in other social and economical terms [12].
(See lines 49-57)